# Adjustment of Micro- and Macroporosity of ß-TCP Scaffolds Using Solid-Stabilized Foams as Bone Replacement

**DOI:** 10.3390/bioengineering10020256

**Published:** 2023-02-15

**Authors:** Lukas Dufner, Bettina Oßwald, Jan Eberspaecher, Bianca Riedel, Chiara Kling, Frank Kern, Michael Seidenstuecker

**Affiliations:** 1Institute for Manufacturing Technologies of Ceramic Components and Composites, University of Stuttgart, 70569 Stuttgart, Germany; 2G.E.R.N. Center of Tissue Replacement, Regeneration & Neogenesis, Department of Orthopedics and Trauma Surgery, Medical Center—Albert-Ludwigs-University of Freiburg, Faculty of Medicine, Albert-Ludwigs-University of Freiburg, Hugstetter Straße 55, 79106 Freiburg, Germany

**Keywords:** capillary suspension, bioceramics, foaming, polymer particles, sacrificial templating

## Abstract

To enable rapid osteointegration in bioceramic implants and to give them osteoinductive properties, scaffolds with defined micro- and macroporosity are required. Pores or pore networks promote the integration of cells into the implant, facilitating the supply of nutrients and the removal of metabolic products. In this paper, scaffolds are created from ß-tricalciumphosphate (ß-TCP) and in a novel way, where both the micro- and macroporosity are adjusted simultaneously by the addition of pore-forming polymer particles. The particles used are 10–40 wt%, spherical polymer particles of polymethylmethacrylate (PMMA) (Ø = 5 µm) and alternatively polymethylsilsesquioxane (PMSQ) (Ø = 2 µm), added in the course of ß-TCP slurry preparation. The arrangement of hydrophobic polymer particles at the interface of air bubbles was incorporated during slurry preparation and foaming of the slurry. The foam structures remain after sintering and lead to the formation of macro-porosity in the scaffolds. Furthermore, decomposition of the polymer particles during thermal debindering results in the formation of an additional network of interconnecting micropores in the stabilizing structures. It is possible to adjust the porosity easily and quickly in a range of 1.2–140 μm with a relatively low organic fraction. The structures thus prepared showed no cytotoxicity nor negative effects on the biocompatibility.

## 1. Introduction

Musculoskeletal disorders are among the most common diseases worldwide. They occur about twice as often as heart problems and also more often than all chronic respiratory diseases. All age groups are affected, with the prevalence of medical problems increasing with age. As society ages, there is a natural increase in chronic diseases, resulting in significant economic consequences for society. Musculoskeletal disorders include a variety of conditions that cause discomfort and pain. Bones, joints as well as muscles and the surrounding structures can be affected [1].

To alleviate some of these complaints, bone substitute materials are needed that enable fast, residue-free and safe osteointegration and conduction to replace bone tissue damaged by osteoporosis or bone cancer, for instance. In addition to the materials, the structural design of the implants must also be precisely adapted to the desired conditions of use. The required structures depend, among other things, on the age, sex and condition of the patient, but also on the mechanical stress on the implant, the diet or the calcium level [2]. Structures in the range of ~10 μm play an important role in biochemical, cellular, and physical catalyst signalling pathways involved in cell differentiation, proliferation, migration and death [3,4]. In the literature, various pore sizes are mentioned that are considered to be optimal [5,6]. Karageorgiou et al. [7] further write that the smallest possible pore size of a scaffold, for a positive regeneration of mineralized bone, can be around 100 µm. In particular, pore sizes in the range of 150 µm to 200 µm are suitable as favourable sizes for good bone ingrowth. However, another study was able to show that bone growth could also be observed in scaffolds with pore sizes between 300 µm and 800 µm, and no significant differences were determined for the different pore sizes [8]. According to our previous work [9], micropores in the range of 0.1–20 μm should be aimed for. For macropores, Karageorgiou et al. [7] suggest a range of 100–1000 μm. However, there are also publications where contradictory statements are made about the optimal sizes of micro- and macropores [10]. The pore size distribution also plays an important role. According to Sánchez-Salcedo et al. [11], more than 60% of the pores should have a size between 150–400 μm, and at least 20% should be smaller than 20 μm. When considering porosity, the degree of open porosity is particularly important, as only the open porosity contributes to the permeability of the scaffolds [1].

The generation of micro- and macroporosity in ceramic components can be achieved in various ways. One common technique is the replica technique, which utilizes an organic template that is infiltrated with ceramic suspension, dried and is then thermally debindered. Synthetic materials such as highly porous polymer sponges, as well as natural raw materials such as wood, corals, etc., are used as templates [12,13,14]. Another opportunity is the direct foaming of slurries using air or inert gases [15,16], partial sintering [17], electrospinning [18], freeze casting [19,20], the utilization of capillary suspensions [21], Pickering emulsion [22,23,24], or extrusion [25,26].

Esslinger et al. [27] used Indirect Solid Freeform Fabrication (iSFF) to influence the porosity in ß-tricalciumphosphate (ß-TCP) scaffolds, which are biocompatible [28,29,30]. By introducing ribs into the thermoplastic moulds used in this process and subsequent debinding, the macroporosity was be selectively adjusted. 

The use of spacer phases as pore formers also enables the adjustment of the pore size, shape and quantity in ceramic components. The spacer particles are added to the ceramic slurries decompose during debinding or are removed by solvents, thus leaving defined voids similar to the replica technique. A wide variety of pore geometries can be realized by selecting the particle shape and size. For instance, salts (NaCl), natural cellulose and synthetic polymers (PA, PVC, PP) are used [12]. In most applications, the pore formers leave only a void corresponding to their negative. By contrast, the method used in this work allows significantly larger structures to be realized with minimal material input.

The possibility of using PMMA particles as placeholder phases to create microporosity in biphasic calciumphosphate scaffolds [31] and its positive effects on cell integration has already been demonstrated by Levengood et al. [32]. In these experiments, macroporosity was created in an additional step via micro-robotic deposition. Parhizkar et al. [33] showed a new technique obtaining PMSQ particles in scaffolds to control porosity.

Biggemann et al. [34] used phenolic resin spheres as placeholder phases for the creation of porous, injection-moulded hydroxyapatite scaffolds. Depending on the former pore content, both open-pore and closed-pore components could be produced. However, foaming of the slurry does not occur in these tests, so the macroporosity can only be influenced indirectly. Gonzenbach et al. [24] already described the hydrophilization of alumina ceramic particles, which leads to foaming of the slurry during slurry preparation and results in the formation of porous ceramics. These are solid-stabilized foams are obtained according to the Pickering emulsion principle.

A key parameter responsible for the solid stabilization of the ceramic foam is the contact angle θ of the particles to the liquid phase. As the hydrophobicity of the particles increases, the contact angle rises, while the compatibility to water decreases. When the particles are dispersed in water and incorporated by stirring, they locate at the interface between the air bubbles and the water (Pickering emulsion). If the air bubbles thus enclosed collide, capillary forces and the liquid film stabilised between the particles prevent the bubbles from coalescing. The structures can remain intact even when liquid is removed [35]. Three-dimensional cross-linked structures are considered to be particularly stable, the formation of which is possible at almost all contact angles. The main determining factors for foam stabilization are particle shape, hydrophobicity, concentration and size. Depending on the particles used, single-layer, multilayer or 3D cross-linked structures can be formed, which differ in their stability [36,37].

In the present work, hydrophobic plastic particles are added to the ceramic slurry during preparation instead of hydrophobic ceramic particles. The foaming of the slurry by the plastic particles creates a network of macropores, while the plastic particles themselves create microporosity.

## 2. Materials and Methods

### 2.1. Production of the Micro- and Macroporous ß-TCP Scaffolds

In order to shape the porous ß-tricalcium phosphate (ß-TCP) scaffolds, small cylindrical moulds were printed by fused deposition modelling (FDM) with a Prusa MK3S (Prusa Research a.s., Prague, Czech Republic) and PLA Silver Filament (3D Jake—niceshops GmbH, Paldau, Austria). The PLA casting moulds (Ø = 8.6 mm, height = 14 mm) produced using FDM are open at the top and bottom and determine the outer geometry of the ceramic scaffolds. The casting slip consisted of ß-TCP raw powder, polymer particles, dispersant and deionized water. The ß-TCP raw powder used (Budenheim, Germany) has a measured D_50_ value of 5.56 μm and an edgy geometry. Spherical polymethylmethacrylates (PMMA), Ø = 5 µm or polymethylsilsesquioxanes (PMSQ), and Ø = 2 µm polymer particles (Coating Products OHZ e.K., Scharmbeck, Germany) with narrow particle size distribution are used as pore-forming agents. Additionally, 1 wt% based on the solids content, Dolapix CE 64 (Zschimmer & Schwarz Chemie GmbH, Lahnstein, Germany), is used as dispersant. A constant solid content of 70 wt% was chosen. This fixed content consisted of various contents of 10, 20, 30 and 40 wt% polymer particles and a corresponding proportion of ceramic particles, see Table 1, samples 1–8. The solid content of 70 wt% was chosen as a result of preliminary experiments, which showed that the slurries prepared at this solid content do not become too viscous. Deionized water makes up for the remaining 30 wt% of the suspension. The individual batch size was constant at 30 g of mixture (solids, water and additives). After premixing, homogenization is carried out in a glass beaker by magnetic stirring at 150 rpm for 60 min to incorporate air and cause foaming of the suspension. Sample 9 represents the control sample containing only ß-TCP without any polymers.

The foamed suspensions were transferred to the polymer moulds placed on a plaster plate using a pipette. This results in liquid extraction from the slip into the plaster plate and the forming of stable green parts. The filled moulds are dried for one day at room temperature. During subsequent de-bindering, the polymer mould and the polymer beads are removed by oxidation (Linn High Therm GmbH, Hirschbach, Germany) at 800 °C. The foamed ß-TCP samples were sintered in air (CeramAix GmbH, Alsdorf, Germany) for 4 h at 1250 °C with a maximum heating rate of 2 K·min^−1^.

### 2.2. Characterisation

For the optical characterization of foam formation, slurries are examined in a Leica DMRP transmission light microscope (Leica Microsystems GmbH, Wetzlar, Germany). Furthermore, green, debindered and pre-sintered parts, as well as sintered parts, are examined by scanning electron microscopy SEM (JEOL Ltd., Akishima, Japan). For SEM, some samples were embedded (EpoThin 2 (Buehler, Leinfeld-Echterdingen, Germany)), ground and polished (Struers, Stuttgart, Germany). The open porosity is measured using the PASCAL 140/440 measuring instruments (Porotec GmbH, Hofheim am Taunus, Germany) according to DIN 66133 based on mercury intrusion. Compressive strength measurements of the sintered specimen are performed following the DIN ISO 17162. A Zwick/Z100 universal testing machine (Zwick Roell GmbH & Co. KG, Ulm, Germany) at a crosshead speed of 5 mm/min and a preload of 0.5 N is used.

### 2.3. Biocompatibility

MG-63 cells (ATCC, CRL 1427) were used for all biocompatibility tests. All tests were performed with 25,000 cells/100 µL per scaffold. Ten identical scaffolds per geometry variation were used per test and all tests were repeated at least 3 times. The biocompatibility tests were carried out exclusively with the mechanically most stable scaffold types.

#### 2.3.1. Live Dead and Hoechst Assay

On each scaffold, 100 µL of the medium was pipetted with 25,000 cells/100 µL of MG-63. The well plates were then incubated for 2 h at 37 °C and a CO_2_ saturation of 5% in an incubator. After two hours, 1 mL of the specific cell medium described previously was added to each well before incubating the well plates in the incubator for 3, 7 and 10 days. The staining solution was prepared by adding 2 mL DPBS (art. no. 14190-094, Gibco, Grand Island, NE, USA) to a Falcon tube (Greiner Bio-One International GmbH, Kremsmünster, Austria) and 4 µL ethidium homodimer III (Eth D-III) solution (together with the calcein part of the Live/Dead Cell Staining Kit II (PromoCell, Heidelberg, Germany)) according to the manufacturer’s protocol (PromoCell, Heidelberg, Germany). An amount of 1 µL of calcein dye was added after mixing the staining solution. All steps were performed in the dark to avoid photobleaching of the staining solution and samples. To eliminate serum esterase activity, all samples at a time point had the medium removed and the cells washed. Staining was then performed according to a previously published protocol [19]. Evaluation was performed using an Olympus fluorescence microscope (BX51, Olympus, Osaka, Japan) at five different positions on the samples at 5× and 10× magnification. In addition to the classic live dead staining, selected samples (one of each group) were also stained with live/dead and Hoechst stains. These differed from the previously described live/dead staining in the dyes used. 

The live staining with Hoechst was prepared by adding 2 mL DPBS (art. no. 14190-094, Gibco, Grand Island, NE, USA) to a Falcon tube (Greiner Bio-One International GmbH, Kremsmünster, Austria), 7.2 µL propidium iodide solution (Sigma Aldrich, now Merck, Darmstadt, Germany) 3.6 µL of calcein dye (art.no. C3099; Thermo Fisher, Waltham, MA, USA) and 36 µL Hoechst (art. No. H1399, Thermo Fisher, Waltham MA, USA) was added after mixing the staining solution. All steps were performed in the dark to avoid photobleaching of staining solution and samples. To eliminate serum esterase activity, all samples at a time point had the medium removed and the cells washed. Evaluation was performed with a Zeiss Colibri 7 Observer fluorescence microscope (Zeiss, Oberkochen, Germany) at five different positions on the samples at 5× and 10× magnification. 

#### 2.3.2. Cell Proliferation (WST-I)

Cells were again seeded on the scaffolds and as control on Thermanox cover slips in the same number and concentration as in the previous biocompatibility tests. After two hours of incubation in the incubator at 37 °C and 5% CO_2_ saturation, the cells adhered to the scaffolds and Thermanox cover slips (as control) so that the respective medium (1 mL) could be added. Plates were then incubated in the incubator for 3, 7 and 10 days. For this purpose, all mediums were aspirated, and all wells were washed three times with PBS. Then, the scaffolds and Thermanox cover slips were transferred to a new 24-well plate. In the old well plate, 300 µL of DMEM medium without phenol red (additives: 1% FBP and 1% P/S) was added to each of the wells where the scaffolds and membranes were previously. In the new well plate, 600 µL of the same medium was added to each of the wells containing the scaffolds and membranes. The blank contains only a DMEM medium without phenol red (same additives) and was measured to account for background absorbance. Finally, 10% WST-I solution was added to all samples of the respective measurement time point (3, 7, 10 days) and incubated for 2 h in the incubator. After 2 h, the absorbance was measured using a spectrometer at 450 nm.

### 2.4. Statistics

All values in this paper are expressed as mean ± standard deviation. The associated calculations were performed using Origin 2020 Professional SR1. ANOVA (Tukey Test) was used for significance testing (*p* < 0.05).

## 3. Results

### 3.1. Foaming Behaviour

Figure 1 shows foamed slurries with 10, 20, 30 and 40 wt% PMSQ beads stirred for 60 min. The polymer content has a decisive influence on the foam stability achieved.

At 10 wt% PMSQ, no stable structures are formed. As soon as stirring is stopped, segregation of the ceramic particles and the rise of previously stabilized air bubbles occurs after a very short time. The shape collapses. Suspensions with 20–40 wt% PMSQ behave differently. The result is a stable, foamed compound that resembles whipped cream, can be shaped as desired, remains stable for hours and maintains its shape even when water is withdrawn. The foamed slurries with 30 and 40 wt% PMSQ stand out in the tests as particularly dimensionally stable. With a high PMSQ content in the slurry, visible foam structures are formed after only a few seconds, but maximum foam stability is achieved after approx. 30 min of stirring. Due to their lower surface ratio, the larger PMMA beads can be incorporated into the slurry much quicker than the fine PMSQ particles.

The foaming behaviour of the PMMA beads with a diameter of 5 μm is significantly weaker (Figure 2). Differences in the flow behaviour of the various slurries can be clearly observed. The higher the polymer content, the more viscous the slurry. Transmitted light microscope images show that foam structures are also formed at least partially in PMMA-stabilised (Figure 3). The large light grey areas represent air bubbles. The polymer beads arrange themselves at the interface between air and liquid, stabilising the liquid film and thus preventing the air bubbles from coalescing.

### 3.2. Drying, Debinding and Sintering

Upon liquid removal, the voids of the air bubbles remain, and the polymer and ceramic particles form a stable structure. In the SEM images of the green parts, partially agglomerated polymer particles can be seen (see Figure 4A). The ceramic particles accumulate primarily in the struts of the foam, confirming the structural information from the transmitted light microscope picture prior to drying (Figure 3). After debinding and sintering, only the ceramic constituents remain (Figure 4B). 

The sharp edges of the samples tend to chip in the debindered as well as in the sintered state, especially with a high degree of foaming of the initial slurry. Despite the high dimensional stability of the foam, shrinkage of the specimens in the order of a few percent occurs during drying. Sintering shrinkage increases with higher polymer content in both PMSQ and PMMA.

The sintered samples produced with different contents of polymer beads look different, see Figure 5. The textures introduced by the addition of PMSQ beads can be classified into microscopic and macroscopic pores. According to Figure 3, increasing the polymer content leads to a stronger foam formation, resulting in a higher macroporosity. As shown in Figure 5 sample SQ4, the sintered structure is fragile and unstable due to the high concentration of macropores and therefore difficult to handle. Areas between the large (≈100 nm) macropores show no significant differences in micropore formation comparing samples with PMSQ and the RAW ß-TCP (Figure 6). In SQ1 and RAW, closed micropores (Figure 5 SQ1) can be observed between the macropores. Increasing PMSQ content leads to an increasing amount of open porosity, clearly visible in Figure 5 SQ3. 

The PMMA samples of various polymer bead fractions macroscopically did not lead to foams as stable as with PMSQ. However, this has an opposed effect on macro- and microporosity, see Figure 7. While less macropores are formed, the micropores in the PMMA-based materials seem more stable and numerous and form a network of open pores, while in the case of PMSQ, more closed pores are observed. A tentative explanation could be that the PMMA completely decomposes, while in the case of PMSQ, a silica residue remains, which enhances sintering and thereby leads to the collapse of micropores [38]. Another important issue is the ratio of TCP particle size to pore size, as smaller pores can be eliminated during sintering while larger pores will remain. The pore of MA resembles an organic structure and seems promising for scaffold materials. With rising PMMA content from 10 wt% to 40 wt%, the open porosity increases. The macropores have an optically rough surface, rougher than the PMSQ samples, due to the porous surrounding matrix.

The results of the mercury porosimetry are shown in Figure 8 and Table 2. For the PMSQ samples, there is a clear correlation. The total porosity and the average pore radius increase with increasing polymer content from sample SQ1 to sample SQ4. For the PMMA samples, there is no evident trend. The highest porosity is shown by sample MA2, the lowest by sample MA1. Sample MA3 and MA4 show the same porosity. Furthermore, the total porosities and the average pore radius of PMMA do not change as much as in PMSQ by increasing the polymer content.

### 3.3. Mechanical Characterisation

The compressive strength of foamed and sintered specimens is relatively low. The measurements were carried out on sintered MA3 samples (*n* = 9). The cylindrical specimens have a diameter of 6.5 mm and a compressive strength of 2.7 ± 0.5 MPa. The samples foamed with PMSQ were too instable for mechanical characterisation. The sharp edges of the samples tend to chip, and the applied pre-load of the test specification already led to the breakage. 

### 3.4. Biocompatibility

#### 3.4.1. Live/Dead Assay

The number of live and dead cells per mm^2^ on the different constructs are shown in Figure 9. With some exceptions, an increase in the number of cells can be seen from day 3 to day 7. After that, stagnation in growth occurred for some samples (SQ3, SQ4, MA1, MA3), and even a decrease (MA2) or increase (MA4, SQ1 and SQ2) in cell numbers from day 7 to day 10 for others. 

Figure 10 below shows an overview of the live/dead staining for the different samples over 10 days. The 3D cell culture control was on Curasan ceramics and the 2D cell culture control on Thermanox cover slips. It is also easy to see the increase in cell numbers over time and the individual differences between samples.

Selected samples were additionally live stained (including Hoechst), shown in Figure 11. The aim of the maximum staining was to counteract the possible intrinsic fluorescence of the ceramic. All cell nuclei glow blue.

#### 3.4.2. Cell Proliferation

Sample SQ2 proliferated over the entire experimental period, whereas samples SQ1 and SQ3 proliferated over 7 days before growth stagnated from day 7 to day 10. The Curasan controls showed a similar value for all three times. In all other samples, the proliferation decreased after day 7 (see Figure 12).

#### 3.4.3. Cytotoxicity

All samples examined showed a cytotoxicity of 0%. Figure 13 clearly shows that all curves in the cells-only (negative control) range are at 0% and partly below, in contrast to the positive control (TritonX (TX)).

## 4. Discussion

### 4.1. Foaming Behaviour

Based on transmission light microscope images of the foamed slurry (Figure 3), the stabilization principle by the combination of Pickering emulsion and capillary effects can be confirmed. In contrast to the described publications [22,23,24,36,37,39], the angular ceramic particles are embedded together with the polymer particles in the inner 3D cross-linked structures, see Figure 14. It should be noted that due to the sample preparation for the transmission light microscope, which involves squeezing the foamed slurry between the carrier glass and the cover plate, the air bubbles deviate from their normally spherical shape. The different foaming behaviour of the PMSQ and PMMA particles used can be explained by their different hydrophobicity and by their different density. However, the probably more important aspect is the different particle size. The smaller PMSQ particles with higher surface to volume ratios are more resistant to incorporation into the slurry, but once incorporated, their larger surface area leads to improved foaming and foam stability. 

According to Horozov et al. [36], as the liquid film becomes thinner, the particles are pulled away from the centre of the film by hydrodynamic forces. This leads to the collapse of the air bubbles. If the suspension contains a higher proportion of particles, denser packed monolayers or bilayers can form, slowing film thinning. Consequently, the lateral mobility of the particles at the film surface has a great influence on the film stability [36,39]. For these reasons, suspensions with low polymer particle content or a high liquid content do not foam completely and tend to decompose. This is the case in this work for all tested PMMA concentrations and the lowest concentration for PMSQ, see Figure 1 and Figure 2. 

Despite the lower density of the polymer particles compared to the ceramic particles, foaming can stabilize structures for a long time without sedimentation. This presumably allows the integration of further materials of different densities into the foam structures. By adding 0.1 wt% Contraspum K 1012, foaming can be prevented or reduced and thus microporosity can be created with priority.

The stabilization of foams with ceramic particles only is not possible without prior hydrophobization of the surfaces as described in [1], since otherwise they do not arrange themselves around air bubbles in the same way as the polymer particles and thus no specific capillary pressure can be built up.

### 4.2. Drying, Debinding and Sintering

If the foam structure is cracked open in the dried state, mainly the inner surfaces of the former air bubbles can be seen. Figure 4 shows only polymer particles around the incorporated air bubbles, which is caused by the hydrophobic character of polymer particles. The oxidative debinding of samples with high polymer content must be performed very carefully with slow debinding rates in order to not damage the microstructure.

Still, at high polymer particle contents, cracks in the scaffolds become visible, which cause a limited stability of these samples. These defects can be explained in several ways. During drying of the foam, a certain volume shrinkage occurs, and stresses can arise due to the adhesion of the foam to the casting mould and uneven drying. Furthermore, thermal expansion of the components occurs during debinding. In particular, the expansion of the polymer beads could lead to damage to the composite, which consists mainly of weakly bonded particles. This bonding is additionally weakened by a high proportion of polymer particles and thus a high degree of foaming. Additional experiments not shown here) showed that the addition of polyvinylalcohole (PVA) binder, higher strength of the green parts, fewer defects during debinding and thus more stable scaffolds can be achieved. However, the exact effect of the PVA content on the foaming behaviour still has to be clarified, so this procedure is not part of this work. 

PMSQ consists of a silicone oxide cage structure bonded to methyl groups [40]. After debinding and sintering, only silica remains. If this residue is not acceptable, another material must be used. PMMA, for example, can be completely removed. 

At first sight, one may expect that the texture of the matrix (microporosity) is the same in both sample types. However, this is not the case. The microporosity of the PMSQ specimens is less pronounced than that of the PMMA samples. Two tentative explanations can be given: Decomposition products of PMSQ (silica) remaining in the material may have an influence on the sintering of the matrix. The dominant effect may, however, be related to the polymer particle/resulting void size. Assuming a particle size of d_50_ = 5.5 µm for ß-TCP and a dense packing this would result in interparticle spaces (octahedric coordination) of 0.414 × 5.5 µm = 2.07 µm. This is exactly the average size of a PMSQ particle. Therefore, the microporosity induced by PMSQ can be removed by sintering, while larger pores introduced by PMMA remain.

### 4.3. Pore Analysis

#### 4.3.1. Macroporosity

The formation of the macropores can be directly related to the foaming behaviour of the slurries. During sample preparation for microscopy, the embedding agent easily penetrated into the scaffolds. This suggests a high degree of accessible open porosity. With increasing polymer content and thus increased foaming, the open porosity increases.

If too small amounts of polymer particles are used, complete stabilization will not be achieved. Due to the incomplete foam stabilization, the resulting rapid segregation of ceramic particles and the rise of air bubbles in the slurry, it is not possible to obtain reproducible results. If the slurry is picked up by pipette after stirring to fill the moulds, segregation already occurs in the pipette. The moulds that are poured first have less porosity, which is the case for PMMA as well as PMSQ beads when the stabilization of the slurry is insufficient. Thus, the measured porosity cannot only be attributed to the polymer content, but also to the preparation method. Nevertheless, it is possible to influence the microstructure and stabilize the entire slip by changing the water content or further increasing the polymer content. However, this was not pursued in this paper. More intense studies are required to determine exactly how properties of the polymer beads such as size, geometry, hydrophobicity and the combination of different beads affect the formation of macropores. According to mercury porosimetry measurements (see Figure 8), the size of the macropores ranges from 5–100 µm in PMMA and PMSQ samples, which is at the lower side of the pore size range of 100 µm defined by [7] for migration and transport. The range between 300–800 µm macropore sizes by [8] was not reached. SEM images indicate that (see Figure 5) the macropore sizes for PMSQ samples are 2 to 3 times higher than size range determined by mercury porosimetry. For PMMA, SEM images (see Figure 7) confirm pore sizes of ~100 µm.

#### 4.3.2. Microporosity

The total porosity and the size distribution of the micropores are determined by mercury intrusion methods. Due to the measuring principle and the existing microstructure, in which macropores are completely surrounded by micropositive structures, it is not possible to provide exact information on the size class distribution with this method. If the size of the inner macropores is to be measured, mercury must first flow through the micropores, which distorts the measurement. The filling of the inner macropores is thus incorrectly classified as microporosity, so the measured values should be viewed with caution. Nevertheless, the method together with the microstructure characterization gives an indication of the size of the microporosity.

An increased polymer content leads to an enlarged space requirement of the polymer particles in the stabilizing bridges, resulting in larger micropores during debinding, which also persist during sintering. 

The dominant fraction of the microporosity [9] ranges between 1 and 20 µm. Some even smaller pores of 0.1 µm were also observed in both PMSQ and PMMA samples. The literature [7,8,9] indicates a favourable pore size of 0.1–300 µm, and this prerequisite is fulfilled for all MA and SQ2–4 materials tested. On the downside, the SQ samples with a very favourable pore size distribution suffer from insufficient mechanical stability. 

### 4.4. Biocompatibility

The biocompatibility tests showed that ß-TCP is very well suited as a biomaterial. No significant differences in biocompatibility were found between the different foaming agents (PMSQ and PMMA), which is not surprising, since the foaming agents burn out in the subsequent sintering process and only porous ß-TCP remains. No negative effect on cytotoxicity was observed. This is not particularly surprising, as ß-TCP has long been considered a good biocompatible biomaterial and has been widely used for the regeneration of bone defects [41,42,43], dental applications [44,45] and drug delivery [46,47]. Although differences in the number of cells per mm^2^ were determined in the live/dead staining, these can be attributed to the different pore sizes and the different hygroscopic properties of the samples. The same applies to the different values in cell proliferation. We have already shown in several previous studies that ß-TCP is very well suited as a bone substitute material [48,49,50]. 

## 5. Conclusions

By adding spherical, hydrophobic PMSQ beads with a diameter of 2 μm or spherical PMMA particles with a diameter of 5 μm, microporosity and macroporosity can be selectively generated in ß-TCP scaffolds. A new feature is the use of polymeric solid particles for the simultaneous generation of macro- as well as microporous scaffold structures. The process combines Pickering emulsion, capillary suspensions, and the simultaneous use of polymer particles as spacer phases for the generation of microporosity. The formation of 3D cross-linked structures results in the foaming of the slurry and the formation of a macroporous, shape-stable foam during processing. This foam can be shaped as desired and remains stable even when dehydrated. 

During subsequent heat treatment, the decomposition of polymeric spacer phases creates the desired microporosity in the remaining ceramic structures. The pore formation is directly influenced by the polymer content used, particle size and hydrophobicity. With the tested particles, porous microstructures with a total porosity of 23–75 vol% and pore diameters of 1.2–140 μm can be obtained. Foaming of the slurries with finer particles is significantly quicker than that with larger particles due to their larger surface area. Only when a certain polymer content is reached is complete foaming of the slurry possible. With increased polymer content, both microporosity and macroporosity increase. By foaming with solid particles, a high porosity with a relatively low organic content can be achieved, which leads to cost advantages and facilitates debinding. Considering the fracture toughness of ß-TCP (0.3–1.1 MPa m^1/2^) [51] and the additional weakening caused by the porous structures, non-load bearing applications are the main target. Overall, the stability and reproducibility of the components must be further improved. The potential of pore formation in ß-TCP by adding polymer particles appears promising.

Biocompatibility of ß-TCP was confirmed by live dead assay and cytotoxicity test and is very well suited as bone substitute material. 

## Figures and Tables

**Figure 1 bioengineering-10-00256-f001:**
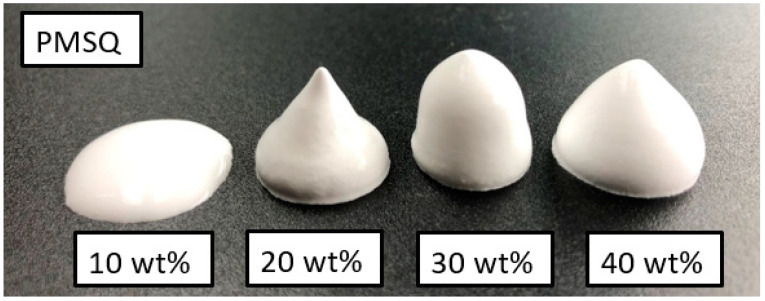
Slurries with 10, 20, 30 and 40 wt% PMSQ after 60 min of magnetic stirring.

**Figure 2 bioengineering-10-00256-f002:**
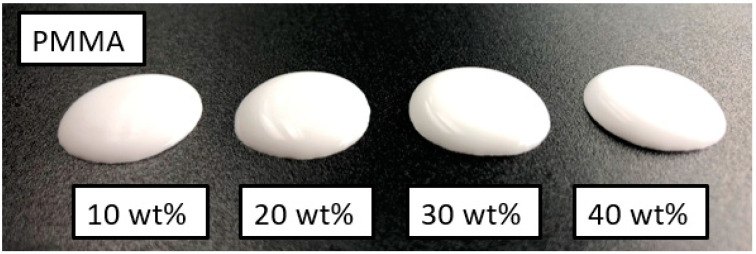
Slurries with 10, 20, 30 and 40 wt% PMMA after 60 min of magnetic stirring.

**Figure 3 bioengineering-10-00256-f003:**
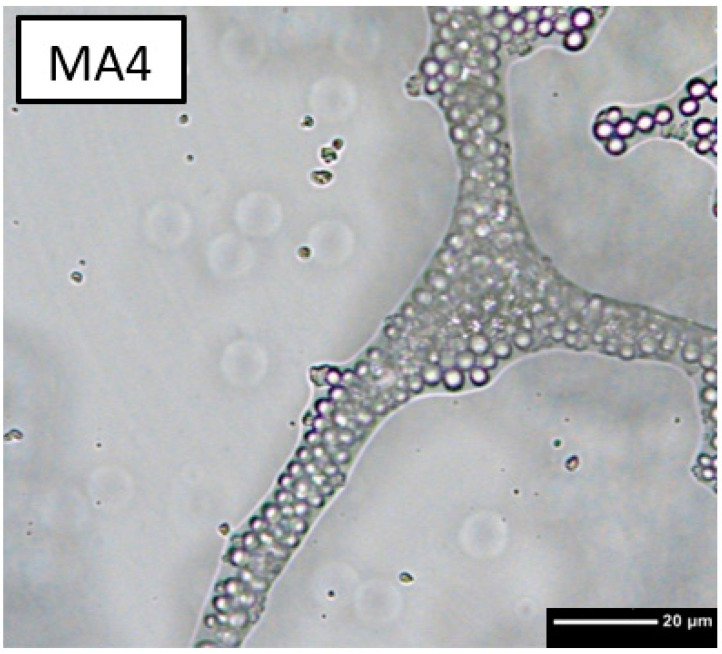
Three-dimnsional crosslinked structures with 40 wt% PMMA and ß-TCP in water under transmission microscope.

**Figure 4 bioengineering-10-00256-f004:**
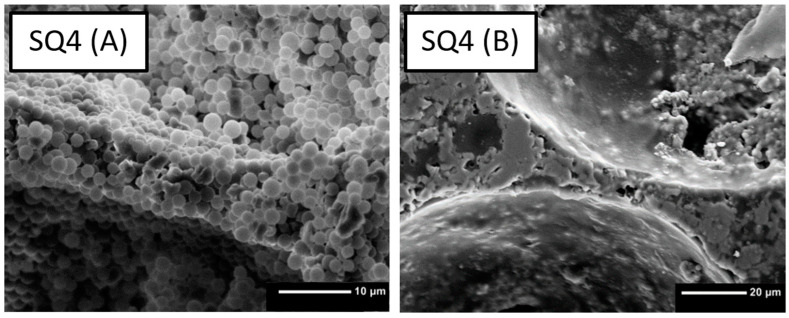
Comparison of non-sintered scaffold (**A**) containing 40 wt% PMSQ with the sintered scaffold (**B**).

**Figure 5 bioengineering-10-00256-f005:**
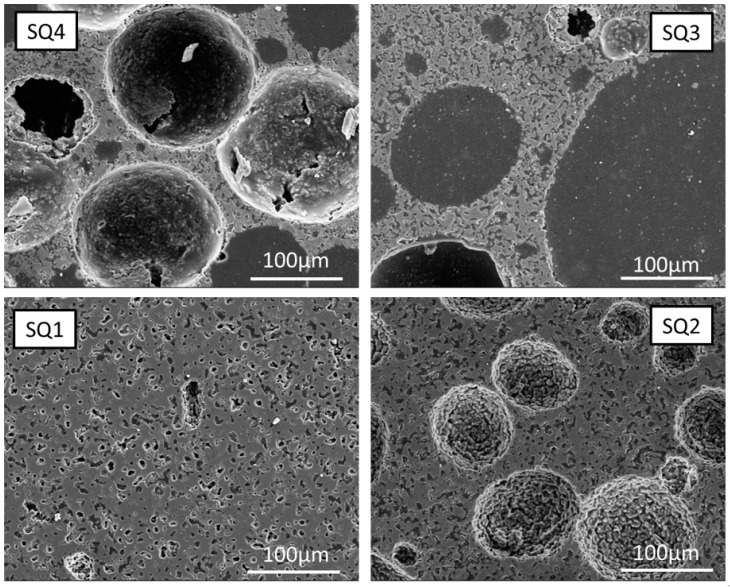
Sintered samples produced with 10–40 wt% PMSQ.

**Figure 6 bioengineering-10-00256-f006:**
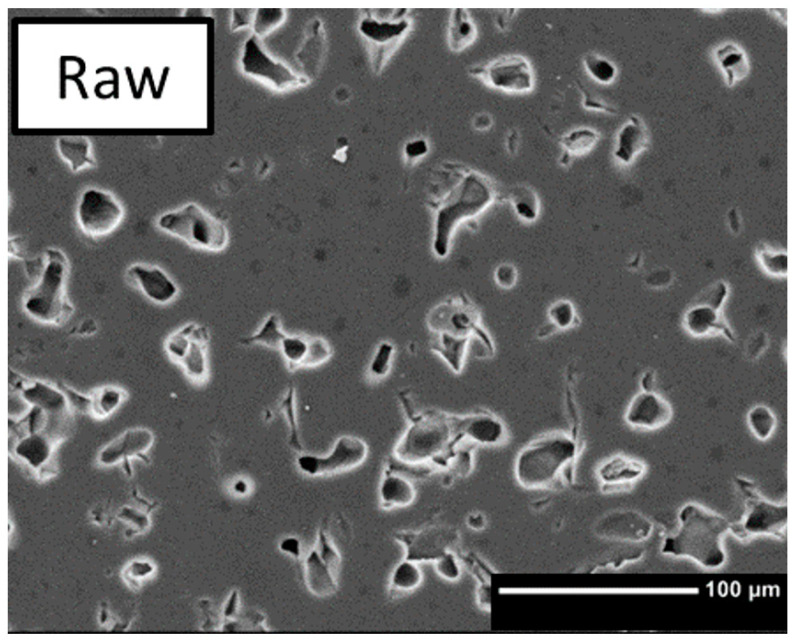
Sintered ß-TCP without polymer beads.

**Figure 7 bioengineering-10-00256-f007:**
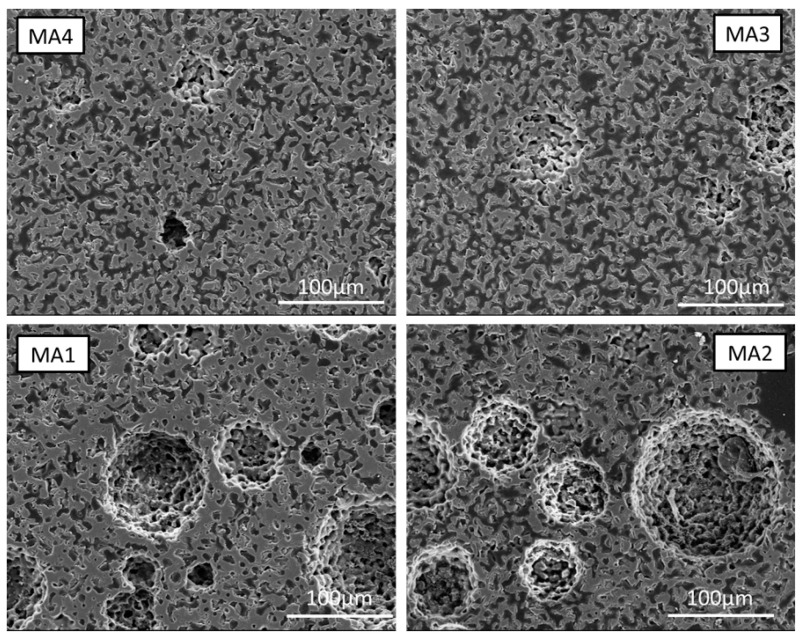
Sintered samples produced with 10–40 wt% PMMA.

**Figure 8 bioengineering-10-00256-f008:**
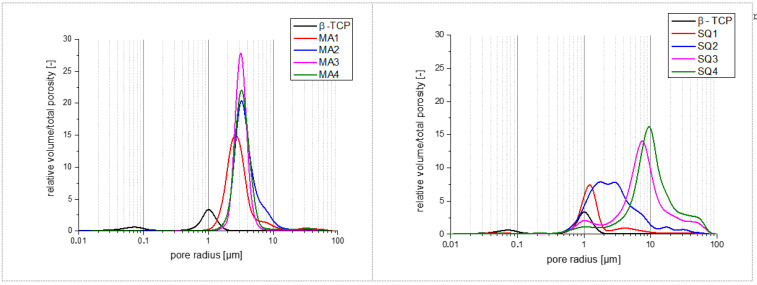
Relative volume and total porosity for PMMA (**left**) and PMSQ (**right**) samples.

**Figure 9 bioengineering-10-00256-f009:**
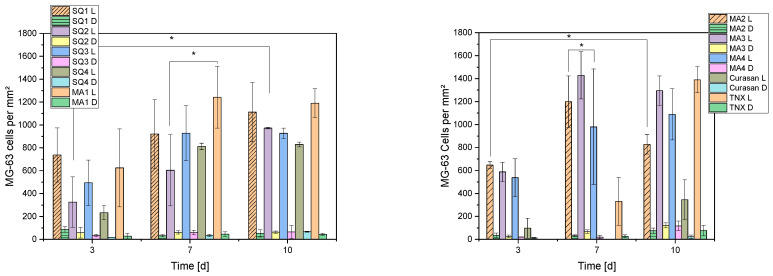
Living and dead cells per mm^2^ for the different compositions. Due to the high number of samples, the presentation was divided into 2 diagrams. L corresponds to living and D to dead cells, TNX is the 2D control on Thermanox cover slips and Curasan the 3D control on ceramics. [*] represents significant difference with *p* < 0.05 (ANOVA).

**Figure 10 bioengineering-10-00256-f010:**
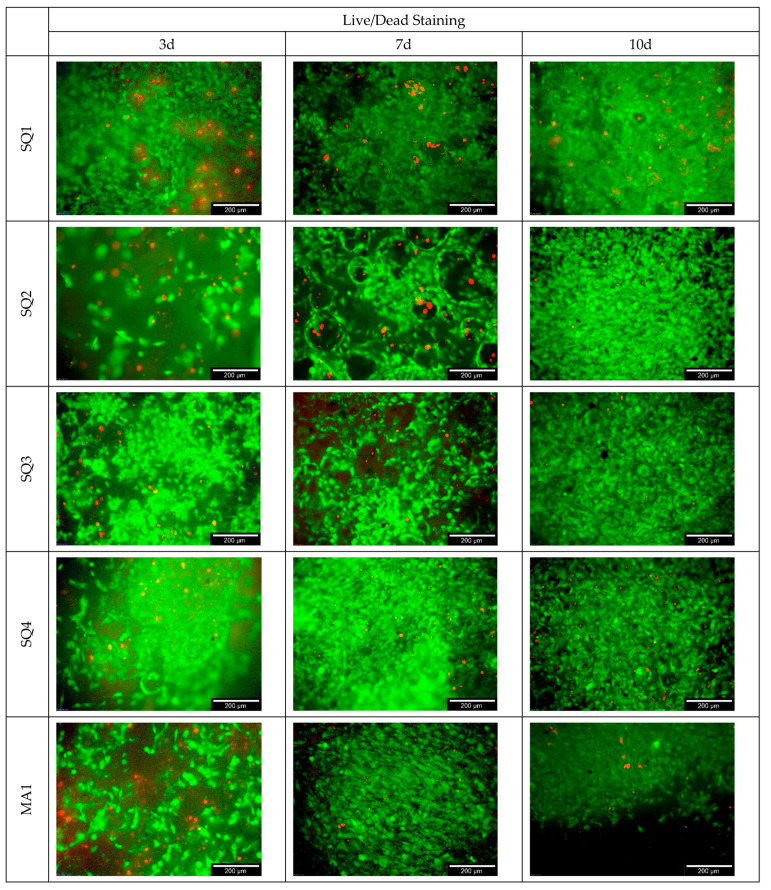
Live/dead staining for the different samples (including the 3D cell culture control on Curasan ceramics as well as 2D cell culture control on Thermanox cover slip).

**Figure 11 bioengineering-10-00256-f011:**
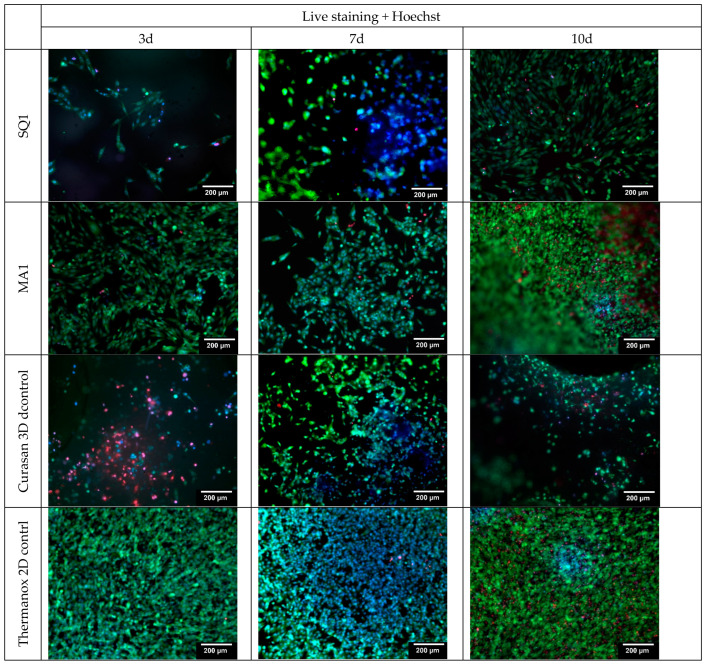
Live/dead and Hoechst staining for the different samples (including the 3D cell culture control on Curasan ceramics as well as 2D cell culture control on Thermanox cover slip); Hoechst (blue) staining of cell core (DNA) of living cells; green = living and red dead cells (live/dead staining).

**Figure 12 bioengineering-10-00256-f012:**
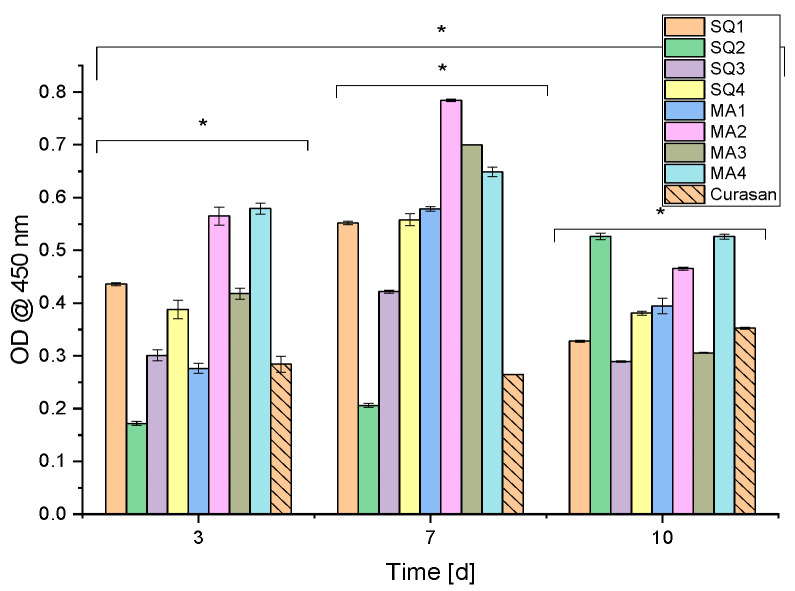
Cell proliferation assay (WST-I): results of the cells on the scaffolds; *n* = 30; with *p* < 0.05 [*].

**Figure 13 bioengineering-10-00256-f013:**
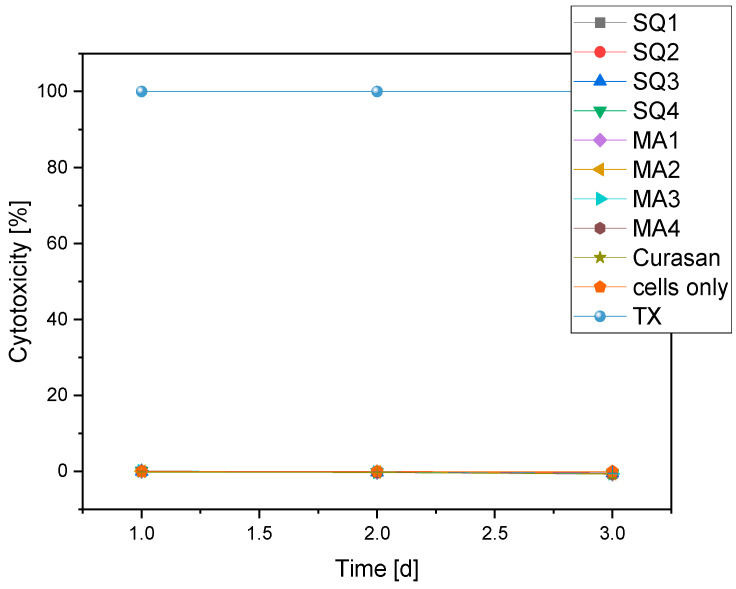
Overview of the cytotoxicity of the different compositions compared to negative (cells only) and positive (TritonX = TX) control.

**Figure 14 bioengineering-10-00256-f014:**
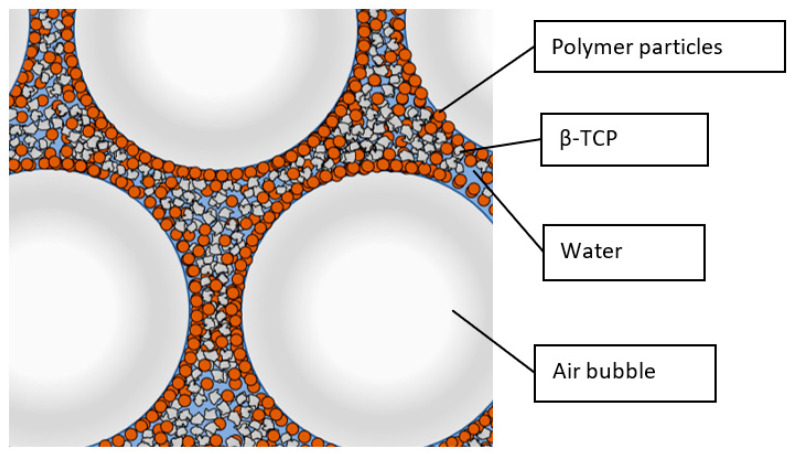
Arrangement of polymer and ceramic particles during solid stabilization.

**Table 1 bioengineering-10-00256-t001:** Slurry composition of the 70 wt% solid content.

No.	Sample	PMSQ [wt%]	PMMA [wt%]	ß-TCP [wt%]
1	SQ1	10	0	90
2	SQ2	20	0	80
3	SQ3	30	0	70
4	SQ4	40	0	60
5	MA1	0	10	90
6	MA2	0	20	80
7	MA3	0	30	70
8	MA4	0	40	60
9	Raw	0	0	100

**Table 2 bioengineering-10-00256-t002:** Mercury porosimetry results of sintered scaffolds.

	SQ1	SQ2	SQ3	SQ4	MA1	MA2	MA3	MA4
Average pore radius [µm]	1.14	2.17	6.68	7.88	2.25	2.59	3.01	2.76
Total porosity [%]	23.2	55.0	68.5	75.2	46.5	55.6	50.1	50.0

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
