# Peer review of "Adjustment of Micro- and Macroporosity of ß-TCP Scaffolds Using Solid-Stabilized Foams as Bone Replacement"

_bioengineering, 2023, doi:10.3390/bioengineering10020256_

Round 1

Reviewer 1 Report

The present work has studied adjustment of micro- and macroporosity of ß-TCP scaffolds using solid-stabilized foams as bone replacement, which has been less discussed before. This work is noteworthy both in terms of scientific and practical novelty. The introduction section has analyzed the literature very well and stated the strengths and weaknesses of the previous works. Also, the purpose of the present work is clearly drawn. The Materials and Methods section has explained in detail the methods of production and characterization of samples. A nice classification has been done for this section. In the results section, the results obtained in the experiments are carefully presented using several figures and graphs, and all the influencing factors during the experiments have been investigated. In addition, in the discussion section, foaming behavior, drying, de-binding, sintering, pore analysis, and biocompatibility have been comprehensively discussed alongside a comparison with relevant works. Finally, in the conclusion section, the most important results obtained in this research are listed. In conclusion, the reviewed paper is very well written and the results are thoughtfully analyzed. I can recommend the article for publication in its present form.

Author Response

Dear reviewer,

thank you for your review. We were very happy about it.

Since you didn't noted anything there was no action required. 

Thank you and best regards! 

Reviewer 2 Report

This paper about the adjustment or creating macro and micro porous in the b-tricalciumphosphate scafolds using solid polymer particles such as PMSQ and PMMA. The subsequent debindering and sintering of scafolds creating macro and microporous to the network.  In addition, the study of cell proliferation, live and dead cell assay are interesting. However, this manuscript need some revision and it can be accepted after major revision

1.  Both PMMA and PMSQ polymers creating pores in different way after debindering or sintering. So author should study the Thermal behaviour of all sample in order to confirm the statement "A tentative explanation could be that the PMMA completely decomposes while in case of PMSQ a silica residue remains which enhances sintering and thereby leads to the collapse of micropores"

2.  Why MA1 was only selected for Hoechst staining study?

3. Provide some more examples for PMMA and PMSQ polymers used in size controlled pore creation. 

Author Response

Dear reviewer,

Thank you for your professional review. Following you will find our answers point by point:

  1. Both PMMA and PMSQ polymers creating pores in different way after debindering or sintering. So author should study the Thermal behaviour of all sample in order to confirm the statement "A tentative explanation could be that the PMMA completely decomposes while in case of PMSQ a silica residue remains which enhances sintering and thereby leads to the collapse of micropores"

You are right. To confirm the statement following citation was added to the text: Thermal degradation of polymethylsilsesquioxane and microstructure of the derived glasses. Citation number 38.

  1. Why MA1 was only selected for Hoechst staining study?

We have chosen one sample of each group for additionally Hoechst staining – but you are right SQ1 was missing. We added the Hoechst staining images for SQ1. We added also more information to the mat & meth section:

“ … In addition to the classic live dead staining, selected samples (one of each group) were also stained with live/dead and Hoechst stains …”

  1. Provide some more examples for PMMA and PMSQ polymers used in size controlled pore creation. 

The following 2 papers were added in the introduction section:

[31] Garcia, R.A.; Tennent, D.J.; Chang, D.; Wenke, J.C.; Sanchez, C.J. An In Vitro Comparison of PMMA and Calcium Sulfate as Carriers for the Local Delivery of Gallium(III) Nitrate to Staphylococcal Infected Surgical Sites. Biomed Res. Int. 2016, 2016, 7078989, doi:10.1155/2016/7078989.

[33] Parhizkar, M.; Sofokleous, P.; Stride, E.; Edirisinghe, M. Novel preparation of controlled porosity particle/fibre loaded scaffolds using a hybrid micro-fluidic and electrohydrodynamic technique. Biofabrication 2014, 6, 45010, doi:10.1088/1758-5082/6/4/045010.

"...The possibility of using PMMA particles as placeholder phases to create microporosity in biphasic calciumphosphate scaffolds [31] and its positive effects on cell integration has already been demonstrated by Levengood et al. [32]. In these experiments, macroporosity was created in an additional step via micro-robotic deposition. Parhizkar et al. [33] showed a new technique obtaining PMSQ particles in scaffolds to control porosity...."

I hope we have been able to answer your questions sufficiently.

Best regards!

Reviewer 3 Report

The scientific paper "Adjustment of micro- and macroporosity of ß-TCP scaffolds using solid-stabilized foams as bone replacement" aimed to create scaffolds from ß-tricalciumphosphate (ß-TCP) and in a novel way, both micro- and macroporosity are adjusted simultaneously by the addition of pore-forming polymer particles.

It can be considered that:

1)      I recommend joining the introductory paragraphs of lines 85-93.

2)      Below figure 1, add the meanings of the abbreviations used in it.

3)      Tab the paragraph at line 179

4)      Reposition figure 3 in the text of the manuscript. The callout in the text must be before inserting the image

5)      In figures 10 and 13, the statistical differences between the analyzed variables were not clearly described

6)      On line 356, adjust to: [19-21,28-30]

7)      I recommend adjusting the discussion. It needs to reference the studies that agree or disagree with the results obtained in the current experiment. For example, in 4.3.1 and 4.3.2 there is no reference.

8)      Divide the conclusion into 2 paragraphs

9)      The manuscript presents a relevant experiment but with few references. I suggest increasing the number of references to give greater scientific support to the results obtained.

Author Response

Dear reviewer,

Thank you for your professional review. Following you will find our answers point by point:

1) I recommend joining the introductory paragraphs of lines 85-93.

It’s done.

2) Below figure 1, add the meanings of the abbreviations used in it

You are right. We updated the caption: Scheme of bubble stabilization by means of hydrophobic solid particles during the drying pro-cess [1] (Air = Air bubble; Water = Media where ceramic particles are suspended; Dry = Drying process, white circles represent air bubbles, black circles represent polymer spheres)

 3) Tab the paragraph at line 179

It’s done.

4) Reposition figure 3 in the text of the manuscript. The callout in the text must be before inserting the image

It’s done.

5) In figures 10 and 13, the statistical differences between the analyzed variables were not clearly described

You are right – we added the [*] for all significant differences (ANOVA) and updated the captions

6) On line 356, adjust to: [19-21,28-30]

It’s done

7) I recommend adjusting the discussion. It needs to reference the studies that agree or disagree with the results obtained in the current experiment. For example, in 4.3.1 and 4.3.2 there is no reference.

You are right. In both sections the pore sizes were discussed and compared with the literature mentioned in the introduction.

For 4.3.1. following text was added: “According to mercury porosimetry measurements (see Figure 9) the size of the macropores ranges from 5-100 µm in PMMA and PMSQ samples which is at the lower side of the pore size range of 100µm defined by [7] for migration and transport. The range between 300-800µm macropore sizes by [8] was not reached. SEM images indicate that (see Figure 6) the macropore sizes for PMSQ samples is 2 to 3 times higher than size range deter-mined by mercury porosimetry. For PMMA SEM images (see Figure 8) confirm pore sizes of ~ 100 µm.”

For 4.3.2. following text was added: “The dominant fraction of the microporosity [9] ranges between 1 and 20µm. Some even smaller pores of 0,1µm were also observed both PMSQ and PMMA samples. Litera-ture [7–9] indicates a favourable pore size of 0.1-300 µm, this prerequisite is fulfilled for all MA and SQ2-4 materials tested. On the downside the SQ samples with a very favoura-ble pore size distribution suffer from insufficient mechanical stability.”

8) Divide the conclusion into 2 paragraphs

The conclusion was divided in 2 paragraphs, line 488, and a part of biocompatibility was added as well, line 502.

“Biocompatibility was confirmed by live-dead assay and cytotoxicity tests, mot sam-ples show an increase of cell proliferation over the first 7 days followed by stagnation.”

9) The manuscript presents a relevant experiment but with few references. I suggest increasing the number of references to give greater scientific support to the results obtained.

You are right, 9 more citations were added:

  • [4] Vesvoranan, O.; Anup, A.; Hixon, K.R. Current Concepts and Methods in Tissue Interface Scaffold Fabrication. Biomimetics (Basel) 2022, 7, doi:10.3390/biomimetics7040151.
  • [5] Raja, N.; Han, S.H.; Cho, M.; Choi, Y.-J.; Jin, Y.-Z.; Park, H.; Lee, J.H.; Yun, H. Effect of porosity and phase com-position in 3D printed calcium phosphate scaffolds on bone tissue regeneration in vivo. Materials & Design 2022, 219, 110819, doi:10.1016/j.matdes.2022.110819.
  • [6] Pecqueux, F.; Tancret, F.; Payraudeau, N.; Bouler, J.M. Influence of microporosity and macroporosity on the mechanical properties of biphasic calcium phosphate bioceramics: Modelling and experiment. Journal of the European Ceramic Society 2010, 30, 819–829, doi:10.1016/j.jeurceramsoc.2009.09.017.
  • [28] Koepp, H.E.; Schorlemmer, S.; Kessler, S.; Brenner, R.E.; Claes, L.; Günther, K.-P.; Ignatius, A.A. Biocompatibility and osseointegration of beta-TCP: histomorphological and biomechanical studies in a weight-bearing sheep model. Biomed. Mater. Res. B Appl. Biomater. 2004, 70, 209–217, doi:10.1002/jbm.b.30034.
  • [29] Lu, L.; Zhang, Q.; Wootton, D.; Chiou, R.; Li, D.; Lu, B.; Lelkes, P.; Zhou, J. Biocompatibility and biodegradation studies of PCL/β-TCP bone tissue scaffold fabricated by structural porogen method. Mater. Sci. Mater. Med. 2012, 23, 2217–2226, doi:10.1007/s10856-012-4695-2.
  • [30] Safronova, T.V.; Selezneva, I.I.; Tikhonova, S.A.; Kiselev, A.S.; Davydova, G.A.; Shatalova, T.B.; Larionov, D.S.; Rau, J.V. Biocompatibility of biphasic α-tricalcium phosphate ceramics in vitro. Mater. 2020, 5, 423–427, doi:10.1016/j.bioactmat.2020.03.007.
  • [31] Garcia, R.A.; Tennent, D.J.; Chang, D.; Wenke, J.C.; Sanchez, C.J. An In Vitro Comparison of PMMA and Calci-um Sulfate as Carriers for the Local Delivery of Gallium(III) Nitrate to Staphylococcal Infected Surgical Sites. Biomed Res. Int. 2016, 2016, 7078989, doi:10.1155/2016/7078989.
  • [33] Parhizkar, M.; Sofokleous, P.; Stride, E.; Edirisinghe, M. Novel preparation of controlled porosity particle/fibre loaded scaffolds using a hybrid micro-fluidic and electrohydrodynamic technique. Biofabrication 2014, 6, 45010, doi:10.1088/1758-5082/6/4/045010.
  • [38] Haußmann, M.; Reznik, B.; Bockhorn, H.; Denev, J.A. Thermal degradation of polymethylsilsesquioxane and microstructure of the derived glasses. Journal of Analytical and Applied Pyrolysis 2011, 91, 224–231, doi:10.1016/j.jaap.2011.02.016.

I hope we have been able to answer your questions sufficiently.

Best regards!

Round 2

Reviewer 2 Report

The manuscript was revised as per the reviewer comments so it can be accepted in present form for publication. 

Reviewer 3 Report

No comments